# External validation of geriatric influenza death score: A multicenter study

**Yuan Kao[1,2]☉, Wei-Jing Lee[1]☉, Kang-Ting Tsai[3,4], Chung-Feng Liu[5], Chien-Chin Hsu[1], Hung-Jung Lin[1,6], Chien-Cheng Huang[1,7]\*, How-Ran Guo[8,9,10]\***

**1** Department of Emergency Medicine, Chi Mei Medical Center, Tainan, Taiwan, **2** Department of Medicine Science Industries, Chang Jung Christian University, Tainan, Taiwan, **3** Department of Senior Services, Southern Taiwan University of Science and Technology, Tainan, Taiwan, **4** Department of Geriatrics and Gerontology, Chi Mei Medical Center, Tainan, Taiwan, **5** Department of Medical Research, Chi Mei Medical Center, Tainan, Taiwan, **6** Department of Emergency Medicine, Taipei Medical University, Taipei, Taiwan, **7** Department of Emergency Medicine, Kaohsiung Medical University, Kaohsiung, Taiwan, **8** Department of Environmental and Occupational Health, College of Medicine, National Cheng Kung University, Tainan, Taiwan, **9** Department of Occupational and Environmental Medicine, National Cheng Kung University Hospital, Tainan, Taiwan, **10** Occupational Safety, Health and Medicine Research Center, National Cheng Kung University Hospital, Tainan, Taiwan

☉ These authors contributed equally to this work.
\* hrguo@mail.ncku.edu.tw (HRG); chienchenghuang@yahoo.com.tw (CCH)

**Data Availability Statement:** Because the data contain potentially sensitive information, derived data supporting the findings of this study are available from the Institutional Review Board of Chi Mei Medical Center on request (contact

## Abstract

The Geriatric Influenza Death (GID) score was developed to help decision making in older patients with influenza in the emergency department (ED), but external validation is unavailable. Thus, we conducted a study was to fill the data gap. We recruited all older patients (≥65 years) who visited the ED of three hospitals between 2009 and 2018. Demographic data and clinical characteristics were retrospectively collected. Discrimination, goodness of fit, and performance of the GID score were evaluated. Of the 5,508 patients (121 died) with influenza, the mean age was 76.6±7.4 (standard deviation) years, and 49.3% were males. The GID score was higher in the mortality group (1.7±1.1 vs. 0.8±0.8, p <0.01). With 0 as the reference, the odds ratio for morality with score of 1, 2 and ≥3 was 3.08 (95% confidence interval [CI]: 1.66–5.71), 6.69 (95% CI: 3.52–12.71), and 23.68 (95% CI: 11.95–46.93), respectively. The area under the curve was 0.722 (95% CI: 0.677–0.766), and the Hosmer–Lemeshow goodness of fit test was 1.000. The GID score had excellent negative predictive values with different cut-offs. The GID score had good external validity, and further studies are warranted for wider application.

## Introduction

Older adults are at high risks of developing serious complications from influenza due to the immune changes with increasing age compared to younger population [1]. Approximately 70%–85% of influenza-related deaths and 50%–70% influenza-related hospitalizations occurred in older population [1]. When older adults are infected with influenza, 67% of them become housebound temporarily and 25% become bedbound temporarily [2]. Therefore, influenza contributed a significant burden to the mortality and morbidity in older population [2].

information: https://www.chimei.org.tw/main/cmh_department/59024/indexInternet.htm).

**Funding:** This study was supported by Grant CMNCKU10816 and Grant CMHCR10954 from Chi Mei Medical Center. The funders had no role in study design, data collection and analysis, decision to publish, or preparation of the manuscript.

**Competing interests:** The authors have declared that no competing interests exist.

Mortality and morbidity in older adults with influenza are high, thus, predicting adverse outcome and early intervention are important. However, clinical manifestations in older adults with influenza are usually atypical and complicated with an underlying illness [3]. A Canadian study recruiting patients aged ≥60 years in the emergency department (ED) revealed that only 31% had temperature of ≥37.8°C and cough, with or without sore throat, which are the criteria for influenza-like illness defined by the United States Centers for Disease Control and Prevention [3]. In 2018, Chung et al. proposed a Geriatric Influenza Death (GID) score, intending to develop a useful clinical decision rule (CDR) to help in decision making for this population [4]. The GID score consists of five predictors as following: severe coma (Glasgow coma scale score of ≤8, 2 points), histories of cancer or coronary artery disease (1 point for each history), elevated C-reactive protein (CRP) levels (>10 mg/dl, 1 point), and bandemia (>10% band cells, 1 point) [4]. The GID score ranges from 0 to 6 [4]. When an older adult is diagnosed with influenza in the ED, the GID score can be calculated easily, and 30-day mortality can be predicted [4]. In a previous study including three risk groups, including low risk group (≤1 point, 1.1% mortality), moderate risk group (2 points, 16.7% mortality), and high risk group (≥3 points, 40% mortality), the area under the curve (AUC) and the Hosmer-Lemeshow goodness of fit tests were 0.861 and 0.578 for the GID score [4]. While the previous study showed that he GID score had good performance in predicting mortality in older patients with influenza in the ED, its validation study is still unavailable. Therefore, we conducted this multicenter study to externally validate the GID score.

## Methods

### Study design, setting, and participants

We conducted a retrospective multicenter study (Chi Mei Medical Center, Chi Mei Liouying Hospital, and Chi Mei Chiali Hospital) and recruited all older patients (≥65 years) who visited these EDs between 2009 and 2018. The study population is Tainan city, which had about 1.8 million of people and 17.3% of them were age ≥65 years until December 31, 2021 [5]. The criteria of influenza were defined as the diagnosis of the International Classification of Diseases, Ninth Revision, Clinical Modification (ICD-9-CM) of 487–488, ICD-10 of J09-J11, or a prescription of Oseltamivir, Peramivir, or Relenza in the index ED visit. We identified patients and collected their demographic data, underlying comorbidities, laboratory data, and mortality were from electronic medical records.

### Definitions of variables

The age was divided into three subgroups: young elderly (65–74 years), moderate elderly (75−84years), and old elderly (≥85years) [4]. The recorded vital signs included Glasgow coma scale (GCS), systolic blood pressure (SBP), heart rate, respiratory rate, and body temperature. Underlying comorbidities included in the study were hypertension, diabetes, chronic obstructive pulmonary disease (COPD), coronary artery disease (CAD), cerebrovascular accident (CVA), malignancy, congestive heart failure (CHF), dementia, and bedridden. Laboratory data included white blood cell count (WBC), bandemia, hemoglobin, platelet, serum creatinine, and high-sensitivity CRP (hs-CRP). Severe coma was defined as GCS ≤ 8 [4]. Bandemia was defined as band >10% [4].

### Measurement of outcome

The outcome was defined as in-hospital mortality. The original measurement of outcome in the GID score is 30-day mortality. Because we were interested in in-hospital mortality, we

chose to use this measurement of outcome in this study. The patients who were discharged from the ED and had no record of mortality in the electronic medical record were also defined as survival.

### Ethical statement

This study was conducted after the approval by the Institutional Review Board of the Chi Mei Medical Center. This is a retrospective study containing de-identified information, and therefore informed consents from patients were waived as it would not affect their rights and welfare.

### Statistics

To evaluate differences in demographic characteristics, underlying comorbidities, and laboratory data between the mortality and the survival groups, we use independent $t$-tests for continuous variables and chi-square tests for categorical variables. Logistic regression analysis was performed for comparing the odds ratio (OR) among patients with different GID scores. The AUC was used to evaluate the discrimination of the score, and the Hosmer–Lemeshow test to evaluate goodness of fit. The performance of GID score was evaluated by sensitivity, specificity, positive predictive value (PPV), and negative predictive value (NPV). The level of significance was set at 0.05 (two-tailed). Because this study is retrospective and missing data is unavoidable in the real world, we chose to give normal values to fill the missing data of Glasgow coma scale, bandemia, and hs-CRP.

### Results

A total of 5,508 patients were identified in this study (Table 1). The mortality rate was 2.2% (121/5508). The mean age (standard deviation) was 76.6 (7.4) years and percentage of males was 49.3%. Patients with mortality predominated in the moderate elderly (41.3%), followed by the old elderly (38.8%), and then the young elderly (19.8%). The average mortality rate was highest in the old elderly (5.2%, 47/902), followed by the moderate elderly (2.2%, 50/2234) and the young elderly (1.0%, 24/2372). Compared to the survival group, the mortality group had lower GCS, SBP, and body temperature but higher heart rate and respiratory rate. Patients with mortality had more underlying comorbidities, including hypertension, COPD, CVA, malignancy, CHF, dementia, and bedridden than patients who survived. As to laboratory data, patients with mortality had higher WBC, bandemia, platelet, serum creatinine, and hs-CRP but lower hemoglobin than those who survived. The average stay in hospital was 7.2 ± 7.5 days. The GID score was higher in patients with mortality than those who survived (1.7 ± 1.1 vs. 0.8 ± 0.8, $p < 0.01$). The missing data and given values are listed in the S1 Table.

Compared to patients with GID score of 0, ORs of mortality in patients with GID score of 1, 2, and ≥3 were 3.08 (95% confidence interval [CI]: 1.66–5.71), 6.69 (95% CI: 3.52–12.71), and 23.68 (95% CI: 11.95–46.93), respectively (Table 2). The AUC was 0.722 (95% CI: 0.677–0.766) (S1 Fig), and the Hosmer-Lemeshow goodness of fit test was 1.000.

In the performance analysis, the NPV of the GID score across three score groups were excellent (0.994 with GID score ≥1, 0.987 with GID score ≥2, and 0.982 with GID score ≥3). For predicting mortality, the GID score cutoff of 1 had the best sensitivity (0.893) and NPV (0.994), and the cutoff of 3 had the best specificity (0.968) (Table 3). Mortality rates associated with GID score of ≥3, GID score of 2, GID score of 1, and GID score of 0 were 13.3%, 4.2%, 1.9%, and 0.6%, respectively (Fig 1).

**Table 1. Characteristics of all older patients with influenza between 2009 and 2018 in ED.**

| Variable | Total patients | Mortality | Survival | p-value |
|---|---|---|---|---|
| | (n = 5508) | (n = 121) | (n = 5387) | |
| Age (years) | 76.6 ± 7.4 | 81.9 ± 8.1 | 76.5 ± 7.4 | <0.01 |
| Age subgroup | | | | |
| Young elderly (65–74) | 2372 (43.1) | 24 (19.8) | 2348 (43.6) | <0.01 |
| Moderately elderly (75–84) | 2234 (40.6) | 50 (41.3) | 2184 (40.5) | |
| Old elderly (≥85) | 902 (16.4) | 47 (38.8) | 855 (15.9) | |
| Sex | | | | |
| Female | 2791 (50.7) | 61 (50.4) | 2730 (50.7) | 0.95 |
| Male | 2717 (49.3) | 60 (49.6) | 2657 (49.3) | |
| Triage vital signs | | | | |
| GCS | 14.5 ± 1.8 | 11.8 ± 3.8 | 14.5 ± 1.7 | <0.01 |
| Severe coma (GCS ≤8) | 160 (2.9) | 31 (25.6) | 129 (2.4) | <0.01 |
| SBP (mm Hg) | 142.9 ± 32.8 | 130.8 ± 33.1 | 143.1 ± 32.7 | <0.01 |
| Heart rate (beats/min) | 93.4 ± 24.2 | 106.4 ± 25.7 | 93.1 ± 24.1 | <0.01 |
| Respiratory rate (breaths/min) | 19.2 ± 3.9 | 22.8 ± 5.2 | 19.1 ± 3.9 | <0.01 |
| Body temperature (˚C) | 37.5 ± 6.6 | 37.1 ± 1.0 | 37.5 ± 6.7 | <0.01 |
| Underlying comorbidity | | | | |
| Hypertension | 3087 (56.1) | 83 (68.6) | 3004 (55.8) | <0.01 |
| Diabetes | 1783 (32.4) | 43 (35.5) | 1740 (32.3) | 0.45 |
| COPD | 709 (12.9) | 37 (30.6) | 672 (12.5) | <0.01 |
| CAD | 1082 (19.6) | 30 (24.8) | 1052 (19.5) | 0.15 |
| CVA | 1034 (18.8) | 46 (38.0) | 988 (18.3) | <0.01 |
| Malignancy | 789 (14.3) | 33 (27.3) | 756 (14.0) | <0.01 |
| CHF | 621 (11.3) | 23 (19.0) | 598 (11.1) | <0.01 |
| Dementia | 585 (10.6) | 39 (32.2) | 546 (10.1) | <0.01 |
| Bedridden | 1759 (31.9) | 96 (79.3) | 1663 (30.9) | <0.01 |
| Laboratory data | | | | |
| WBC (cells/mm$^3$) | 8.5 ± 4.1 | 12.0 ± 6.6 | 8.4 ± 4.0 | <0.01 |
| Band form (%) | 0.2 ± 1.5 | 1.8 ± 4.7 | 0.2 ± 1.4 | <0.01 |
| Bandemia (>10% band), % | 26 (0.5) | 7 (5.8) | 19 (0.4) | <0.01 |
| Hemoglobin (mg/dL) | 12.4 ± 1.9 | 11.6 ± 2.5 | 12.4 ± 1.8 | <0.01 |
| Platelet (10$^3$/mm$^3$) | 187.4 ± 72.4 | 212.7 ± 101.2 | 186.7 ± 71.4 | <0.01 |
| Serum creatinine (mg/dL) | 1.3 ± 1.5 | 1.6 ± 1.4 | 1.3 ± 1.5 | 0.02 |
| hs-CRP (mg/dL) | 28.4 ± 47.7 | 59.3 ± 86.7 | 27.7 ± 46.2 | <0.01 |
| hs-CRP >10 mg/dL, % | 2531 (46.0) | 71 (58.7) | 2460 (45.7) | <0.01 |
| GID score | 0.9 ± 0.8 | 1.7 ± 1.1 | 0.8 ± 0.8 | <0.01 |
| Mortality | 121 (2.2) | | | |

Data were presented as n (%) or mean ± SD. ED, Emergency Department; SD, standard deviation; GCS, Glasgow coma scale; SBP, systolic blood pressure; COPD, chronic obstructive pulmonary disease; CAD, coronary artery disease; CVA, cerebrovascular accident; CHF, congestive heart failure; WBC, white blood cell count; hs-CRP, high sensitivity C-reactive protein; GID, geriatric influenza death.

## Discussion

This external validation study showed that GID score had an acceptable discrimination and a good fit for predicting mortality for older patients with influenza in the ED. The mortality rate increased with the GID score. The GID score performed best in NPV. GID score of ≥1 had

**Table 2. Logistic regression analysis, AUC, and Hosmer–Lemeshow goodness of fit test for validating the accuracy of GID score in older patients with influenza in ED.**

| Variable | B | Odds ratio | $p$-value | 95% CI |
|---|---|---|---|---|
| GID score of 0 | – | 1 (reference) | – | – |
| GID score of 1 | 1.13 | 3.08 | <0.01 | 1.66–5.71 |
| GID score of 2 | 1.90 | 6.69 | <0.01 | 3.52–12.71 |
| GID score of ≥3 | 3.17 | 23.68 | <0.01 | 11.95–46.93 |
| AUC | 0.722 | | | 0.677–0.766 |
| Hosmer-Lemeshow goodness of fit test | 1.000 | | | |

CI, confidence interval; AUC, area under the curve; GID, geriatric influenza death; ED, Emergency Department.

**Table 3. Performance of GID score in predicting mortality in older patients with influenza in ED.**

| | GID score of ≥1 | GID score of ≥2 | GID score of ≥3 |
|---|---|---|---|
| Sensitivity | 0.893 | 0.504 | 0.215 |
| Specificity | 0.376 | 0.817 | 0.968 |
| PPV | 0.031 | 0.058 | 0.132 |
| NPV | 0.994 | 0.987 | 0.982 |

Data were presented as %. GID, geriatric influenza death; ED, Emergency Department; positive predictive value; negative predictive value.

the best sensitivity and NPV for predicting mortality, whereas GID score of ≥3 had the best specificity.

Compared to the original study that developed the GID score, the mortality rate in this validation study was lower (2.2% vs. 4.9%). Possible reasons are different population and

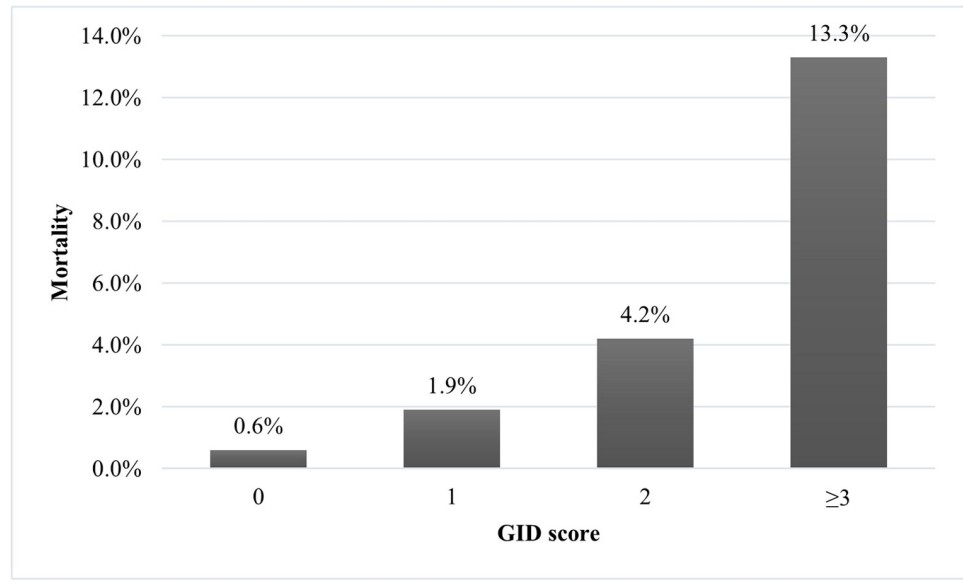

**Fig 1. Mortality rates in three groups of GID score (0 vs. 1 vs. 2 vs. ≥3).** GID, geriatric influenza death.

measurement of mortality. The original GID score was developed in an ED of a medical center in northern Taiwan [4], whereas the validation study was conducted in three hospitals in southern Taiwan. In addition, the validation study recruited 5,508 patients, which is a far larger sample size than in the original study (409 patients). We adopted in-hospital mortality in the validation study because of the difficulty in following 5,508 patients after discharge, which is also different from the original study. The "in-hospital mortality" should be lower than the "30-day mortality" due to the shorter follow-up period. In the literature, the mortality in older adults with influenza has a great variation, ranging from 0.009% to 14.3% in the nursing home for the elderly [6] due to different race, medical care system, and time period.

The AUC in the validation study is 0.722, lower than the 0.861 AUC in the original study [4]. Other studies have revealed that CDRs always perform better in the dataset which they derived than their applications either in internal or external validations [6, 7]. Lower performance in the validation study may be due to differences in patient samples and prevalence of the disease, over-fitting, unsatisfactory model derivation, absence of important predictors, and differences in interpretation and measurement of predictors [8, 9]. Even if a CDR is well developed, it is not necessarily be generalizable to new populations [8]. Thus, external validation, such as this study, is an essential process to assess the performance of a CDR [8].

Aims of external validation includes taking the original model and its predictors and regression coefficients, measuring the predictor and outcome variables in a new population, applying the original CDR to these data to predict the outcome of interest, and comparing the predictive performance of the CDR by analyzing outcomes [8, 10]. The evaluation of a CDR performance can be done by discrimination, calibration, and measurement to quantify clinical usefulness such as decision curve analysis [8, 11]. A CDR can be revised according to results if it performs poorly in the external validation [8]. Despite external validation as an essential process for clinical use of CDR, few CDRs have been externally validated [8, 12–14]. In a systematic review of 101 CDRs showed that 76% of them had no such validation, 17% had narrow validation, 8% had broad validation, and none had impact analysis [14].

The GID score performs well for the NPV across different scores in this study. In addition to the well-developed GID score, another explanation may be due to low mortality rate in the study population, which also explains the low PPV (0.031 with the GID score cut-off of 1, 0.058 with the cut-off of 2, and 0.132 with the cut-off of 3) in this study. The GID score cut-off of 1 had a sensitivity of 0.893, which suggests that the cut-off "1" may be used for identifying patients with a high mortality risk. The GID score cut-off of 3 had a specificity of 0.968, which suggests that the cut-off "3" may be used for identifying patients with a low mortality risk. The cut-off point depends on the aim of clinicians.

In the development and evaluation of CDR, three main stages, including derivation, validation, and impact analysis were found [8, 15]. Suggested requirements for designing external validation include a prospective multicenter design, a minimum of 100 outcome events, and a framework of generalizability to enhance the interpretation of findings [8, 16]. Suggested requirements for types of external validation are conducting temporal, geographical, and domain validation studies and meta-analysis using a published framework to summarize the overall performance of the CDR [8, 17]. After the validation, refinement of a CDR, including model updating or adjustment are suggested [8]. Final aim of a CDR is to improve the quality of care, thus, the impact of a validated CDR on patient outcomes and clinician behavior is suggested to be examined [8, 18]. Evaluation of cost-effectiveness and long-term implementation and dissemination are also advised after the impact analysis [8].

The major strength of this study is that it is the first external validation study for GID score. The study had a large-sample size and used multicenter design. There are some limitations as the follows. First, there were different epidemics within the time range that could have biased

the results. The major circulating viruses were influenza A (H1N1) in 2009, 2010, 2015, and 2018; influenza A (H3N2) in 2012, 2013, 2014, and 2016; and influenza B in 2011 and 2017 [19]. Second, the definition of disease in the previous paper was presence of fever and identification of flu in nasal swabs whereas in this case it is influenza diagnosis or prescription of common anti-flu drugs. The measurement of outcome was in-hospital mortality in this study, which is different from 30-day mortality in the original study for developing GID score. The difference of disease definition and outcome measurement between two studies may contributed to a lower accuracy in this study. However, the result of this study for predicting in-hospital mortality could provide an important reference for validating GID score for different outcome. Because we had no data about the percentage of death within 30 days after discharge, further study about this issue is needed. Before the widely use of the GID score, more validation studies, effects of epidemics of different viruses, refinement, and preparation for impact analysis may be warranted.

## Conclusions

This multicenter study assessed the external validity of the GID score. The discrimination is acceptable, and the fit is good. The GID score has the best performance for NPV. The GID score cut-off of 1 is suggested to be used for identifying patients with a high mortality risk, and the cut-off 3 is suggested to be used for identifying patients with a low mortality risk. Further studies, including more validation studies, refinement, and preparation for impact analysis are suggested.

## Supporting information

**S1 Fig. The AUC of GID score.** AUC, area under the curve; GID, Geriatric Influenza Death. (DOCX)

**S1 Table. Missing data and their given values.**
(DOCX)

## Acknowledgments

We thank Ms. Chia-Jung Chen and Ms. Tzu-Lan Liu for their assistance in data collection and management and Ms. Yu-Shan Ma for the assistance with statistics. We thank Enago for the English revision.

## Author Contributions

**Conceptualization:** Yuan Kao, Wei-Jing Lee, Chien-Cheng Huang, How-Ran Guo.

**Data curation:** Chung-Feng Liu.

**Formal analysis:** Chung-Feng Liu.

**Funding acquisition:** Chien-Chin Hsu, How-Ran Guo.

**Investigation:** Yuan Kao, Wei-Jing Lee, Chung-Feng Liu, Chien-Cheng Huang, How-Ran Guo.

**Methodology:** Yuan Kao, Wei-Jing Lee, Chung-Feng Liu, Chien-Cheng Huang, How-Ran Guo.

**Project administration:** Yuan Kao, Wei-Jing Lee, Kang-Ting Tsai, Chung-Feng Liu, Chien-Chin Hsu, Hung-Jung Lin, Chien-Cheng Huang, How-Ran Guo.

**Resources:** Chung-Feng Liu.

**Software:** Chung-Feng Liu.

**Supervision:** Chien-Cheng Huang, How-Ran Guo.

**Validation:** Chien-Cheng Huang.

**Visualization:** Chien-Cheng Huang.

**Writing – original draft:** Yuan Kao, Wei-Jing Lee, Chien-Cheng Huang.

**Writing – review & editing:** Yuan Kao, Wei-Jing Lee, Kang-Ting Tsai, Chung-Feng Liu, Chien-Chin Hsu, Hung-Jung Lin, Chien-Cheng Huang, How-Ran Guo.

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
