## [Decision Letter · Decision Letter 0]

21 Jul 2022

PONE-D-22-17460External validation of geriatric influenza death score: a multicenter studyPLOS ONE

Dear Dr. Huang,

Thank you for submitting your manuscript to PLOS ONE. After careful consideration, we feel that it has merit but does not fully meet PLOS ONE’s publication criteria as it currently stands. Therefore, we invite you to submit a revised version of the manuscript that addresses the points raised during the review process.

We look forward to receiving your revised manuscript.

Kind regards,

Muhammad Tarek Abdel Ghafar, M.D

Academic Editor

PLOS ONE

Journal Requirements:

"This study was supported by Grant CMNCKU10816 and Grant CMHCR10954 from Chi Mei Medical Center."

"NO. The funders had no role in study design, data collection and analysis, decision to publish, or preparation of the manuscript."

Reviewers' comments:

Reviewer's Responses to Questions

**Comments to the Author**

1. Is the manuscript technically sound, and do the data support the conclusions?

Reviewer #1: Yes

Reviewer #2: Yes

2. Has the statistical analysis been performed appropriately and rigorously? 

Reviewer #1: Yes

Reviewer #2: Yes

3. Have the authors made all data underlying the findings in their manuscript fully available?

Reviewer #1: Yes

Reviewer #2: Yes

4. Is the manuscript presented in an intelligible fashion and written in standard English?

Reviewer #1: Yes

Reviewer #2: Yes

5. Review Comments to the Author

Reviewer #1: Please correct the typos

Please describe the population so that one gets a feel of comorbidity. What was the average stay in hospital? what % died within 30 days but after discharge in previous studies

comparing percentages of the outcome does not give a correct position of reality. for example 38%of deaths were among the old elderly but the denominator is smaller than the other groups

Being retrospective- did you have any missing data?

Reviewer #2: The original sent for my review is a retrospective validation of a risk prediction rule for identifying older patients at risk of mortality from flu. The authors involved in the first description are now involved in this external validation

The results of their work showed that this GID does identify patients at a higher risk of mortality from flu, showing a lower mortality rates for different cut offs compared to those previously published.

The work is well done but need some clarifications by the authors to improve its overall quality.

- The main concerns come from the definition of disease (in the previous paper was fever+ identification of flu in nasal swabs whereas in this case it is influenza diagnosis or prescription of common anti flu drugs) and the different outcomes (which is recognized by the authors as a limitation). However, how can they be confident that those patients who went to the ED and discharged to home did not die? There is no reason to use all those who did not die in the hospital or were discharged to be supposed not to die.

- Another concern comes from the different epidemics within the time range that could have biased the results (more aggressive variants, different vaccination %). Could the authors provide some data regarding this concern?

6. PLOS authors have the option to publish the peer review history of their article (what does this mean?). If published, this will include your full peer review and any attached files.

Reviewer #1: **Yes: **Evans Amukoye

Reviewer #2: **Yes: **Bernardino Alcazar

---

## [Author Response · Author response to Decision Letter 0]

20 Jan 2023

Response to the Reviewers’ comments (Manuscript ID: PONE-D-22-17460)

Thank you very much for giving us the opportunity to revise the manuscript for your reconsideration of publication in the PLOS ONE. According to the valuable comments and suggestions from the reviewer, we revised the manuscript and indicated the changes in red and line number. In addition, we prepared a “Response to Reviewers’ Comments” which includes point-by-point responses to each reviewer as the follows.

Editorial office

Response: We have revised the manuscript according to your suggestions. 

Response: We have ensured the grant information we provided in the ‘Funding Information’ and ‘Financial Disclosure’ sections does match in the revision. 

3. Thank you for stating the following in the Acknowledgments Section of your manuscript: "This study was supported by Grant CMNCKU10816 and Grant CMHCR10954 from Chi Mei Medical Center." We note that you have provided funding information that is not currently declared in your Funding Statement. However, funding information should not appear in the Acknowledgments section or other areas of your manuscript. We will only publish funding information present in the Funding Statement section of the online submission form. Please remove any funding-related text from the manuscript and let us know how you would like to update your Funding Statement. Currently, your Funding Statement reads as follows: "NO. The funders had no role in study design, data collection and analysis, decision to publish, or preparation of the manuscript." Please include your amended statements within your cover letter; we will change the online submission form on your behalf.

Response: We have deleted the funding in the manuscript and added funding statement “This study was supported by Grant CMNCKU10816 and Grant CMHCR10954 from Chi Mei Medical Center. The funders had no role in study design, data collection and analysis, decision to publish, or preparation of the manuscript.” to the cover letter according to your suggestion. 

4. We note that you have indicated that data from this study are available upon request. PLOS only allows data to be available upon request if there are legal or ethical restrictions on sharing data publicly. For more information on unacceptable data access restrictions, please see http://journals.plos.org/plosone/s/data-availability#loc-unacceptable-data-access-restrictions. In your revised cover letter, please address the following prompts: a) If there are ethical or legal restrictions on sharing a de-identified data set, please explain them in detail (e.g., data contain potentially sensitive information, data are owned by a third-party organization, etc.) and who has imposed them (e.g., an ethics committee). Please also provide contact information for a data access committee, ethics committee, or other institutional body to which data requests may be sent. b) If there are no restrictions, please upload the minimal anonymized data set necessary to replicate your study findings as either Supporting Information files or to a stable, public repository and provide us with the relevant URLs, DOIs, or accession numbers. For a list of acceptable repositories, please see http://journals.plos.org/plosone/s/data-availability#loc-recommended-repositories. We will update your Data Availability statement on your behalf to reflect the information you provide.

Response: We have revised the Availability of data and materials as “Because the data contain potentially sensitive information, derived data supporting the findings of this study are available from the Institutional Review Board of Chi Mei Medical Center on request (contact information: https://www.chimei.org.tw/main/cmh_department/59024/indexInternet.htm).” In addition, the above statement was added to the cover letter. 

Reviewer #1: 

1. Please correct the typos.

Response: We thank the Reviewer’s reminding and have checked and corrected the typos through the whole manuscript. 

2. Please describe the population so that one gets a feel of comorbidity. What was the average stay in hospital? what % died within 30 days but after discharge in previous studies comparing percentages of the outcome does not give a correct position of reality. for example 38%of deaths were among the old elderly but the denominator is smaller than the other groups.

Response: We thank the Reviewer’s comment regarding describing the study population and added “The study population is Tainan city, which had about 1.8 million of people and 17.3% of them were age ≥65 years until December 31, 2021 [5].” to describe the population in the Methods of the revised manuscript (line 93-95). 

We added “The average stay in hospital was 7.2 ± 7.5 days.” into the Results of the revised manuscript (line 150-151). 

Because we were interested in in-hospital mortality in this study, we did not include 30-day mortality in all patients, including the patients who were discharged. To explain the reason, we added “The original measurement of outcome in the GID score is 30-day mortality. Because we were interested in in-hospital mortality, we chose to use this measurement of outcome in this study. The patients who were discharged from the ED and had no record of mortality in the electronic medical record were also defined as survival.” to the Measurement of outcome (line 115-119) and “The measurement of outcome was in-hospital mortality in this study, which is different from 30-day mortality in the original study for developing GID score. The difference of disease definition and outcome measurement between two studies may contributed to a lower accuracy in this study. However, the result of this study for predicting in-hospital mortality could provide an important reference for validating GID score for different outcome. Because we had no data about the percentage of death within 30 days after discharge, further study about this issue is needed.” to the limitations of the revised manuscript (line 239-245). 

We agree with the Reviewer’s comment regarding that outcome does not give a correct position of reality. Therefore, we added “The average mortality rate was highest in the old elderly (5.2%, 47/902), followed by the moderate elderly (2.2%, 50/2234) and the young elderly (1.0%, 24/2372).” to the Results of the revised manuscript to make it clearer (line 145-147). 

3. Being retrospective- did you have any missing data?

Response: We thank the Reviewer’s comment regarding the missing data and added “Because this study is retrospective and missing data is unavoidable in the real world, we chose to give normal values to fill the missing data of Glasgow coma scale, bandemia, and hs-CRP.” to the Statistics (line 136-138) and “The missing data and given values are listed in the Supplementary Table 1.” to the Results of the revised manuscript (line 154-155). 

Reviewer #2: 

1. The main concerns come from the definition of disease (in the previous paper was fever+ identification of flu in nasal swabs whereas in this case it is influenza diagnosis or prescription of common anti flu drugs) and the different outcomes (which is recognized by the authors as a limitation). However, how can they be confident that those patients who went to the ED and discharged to home did not die? There is no reason to use all those who did not die in the hospital or were discharged to be supposed not to die.

Response: We agreed with the Reviewer’s comment regarding the definition of disease and the different outcomes and added “Second, the definition of disease in the previous paper was presence of fever and identification of flu in nasal swabs whereas in this case it is influenza diagnosis or prescription of common anti-flu drugs. The measurement of outcome was in-hospital mortality in this study, which is different from 30-day mortality in the original study for developing GID score. The difference of disease definition and outcome measurement between two studies may contributed to a lower accuracy in this study. However, the result of this study for predicting in-hospital mortality could provide an important reference for validating GID score for different outcome. Because we had no data about the percentage of death within 30 days after discharge, further study about this issue is needed.” to the limitations of the revised manuscript (line 235-244). In addition, we added “The original measurement of outcome in the GID score is 30-day mortality. Because we were interested in in-hospital mortality, we chose to use this measurement of outcome in this study.” to the Measurement of outcome to explain why we chose in-hospital mortality as the outcome in this study (line 115-117). To response to another concern for the definition of mortality, we added “The patients who were discharged from the ED and had no record of mortality in the electronic medical record were also defined as survival.” to the Measurement of outcome of the revised manuscript (line 117-119). 

2. Another concern comes from the different epidemics within the time range that could have biased the results (more aggressive variants, different vaccination %). Could the authors provide some data regarding this concern?

Response: We thank the Reviewer’s comment regarding different epidemics within the time range that could have biased the results. We revised the sentences as “First, there were different epidemics within the time range that could have biased the results. The major circulating viruses were influenza A (H1N1) in 2009, 2010, 2015, and 2018; influenza A (H3N2) in 2012, 2013, 2014, and 2016; and influenza B in 2011 and 2017 [19].” in the limitations of the revised manuscript (line 232-235).

---

## [Decision Letter · Decision Letter 1]

9 Mar 2023

External validation of geriatric influenza death score: a multicenter study

PONE-D-22-17460R1

Dear Dr. Huang,

We’re pleased to inform you that your manuscript has been judged scientifically suitable for publication and will be formally accepted for publication once it meets all outstanding technical requirements.

Kind regards,

Muhammad Tarek Abdel Ghafar, M.D

Academic Editor

PLOS ONE

Additional Editor Comments (optional):

Reviewers' comments:

Reviewer's Responses to Questions

**Comments to the Author**

1. If the authors have adequately addressed your comments raised in a previous round of review and you feel that this manuscript is now acceptable for publication, you may indicate that here to bypass the “Comments to the Author” section, enter your conflict of interest statement in the “Confidential to Editor” section, and submit your "Accept" recommendation.

Reviewer #1: All comments have been addressed

Reviewer #2: All comments have been addressed

2. Is the manuscript technically sound, and do the data support the conclusions?

Reviewer #1: Yes

Reviewer #2: Yes

3. Has the statistical analysis been performed appropriately and rigorously? 

Reviewer #1: Yes

Reviewer #2: Yes

4. Have the authors made all data underlying the findings in their manuscript fully available?

Reviewer #1: Yes

Reviewer #2: Yes

5. Is the manuscript presented in an intelligible fashion and written in standard English?

Reviewer #1: Yes

Reviewer #2: Yes

6. Review Comments to the Author

Reviewer #1: I am comfortable with authors response. This tool may require further validations before recommending wide use.

Reviewer #2: The authors have assessed the main concerns from this reviewer properly. They gave reasonable explanations of the outcome assessment and posible biases from the study

7. PLOS authors have the option to publish the peer review history of their article (what does this mean?). If published, this will include your full peer review and any attached files.

Reviewer #1: No

Reviewer #2: **Yes: **Bernardino Alcazar-Navarrete

---

## [Editor Report · Acceptance letter]

16 Mar 2023

PONE-D-22-17460R1 

External validation of geriatric influenza death score: a multicenter study 

Dear Dr. Huang:

I'm pleased to inform you that your manuscript has been deemed suitable for publication in PLOS ONE. Congratulations! Your manuscript is now with our production department. 

Kind regards, 

on behalf of

Prof Muhammad Tarek Abdel Ghafar 

Academic Editor

PLOS ONE